# The two types of society: Computationally revealing recurrent social formations and their evolutionary trajectories

**Lux Miranda**(iD)*, **Jacob Freeman**(iD)

Anthropology Program, Utah State University, Logan, Utah, United States of America

* lux@luxmiranda.com

## Abstract

Comparative social science has a long history of attempts to classify societies and cultures in terms of shared characteristics. However, only recently has it become feasible to conduct quantitative analysis of large historical datasets to mathematically approach the study of social complexity and classify shared societal characteristics. Such methods have the potential to identify recurrent social formations in human societies and contribute to social evolutionary theory. However, in order to achieve this potential, repeated studies are needed to assess the robustness of results to changing methods and data sets. Using an improved derivative of the *Seshat: Global History Databank*, we perform a clustering analysis of 271 past societies from sampling points across the globe to study plausible categorizations inherent in the data. Analysis indicates that the best fit to Seshat data is five subclusters existing as part of two clearly delineated superclusters (that is, two broad "types" of society in terms of social-ecological configuration). Our results add weight to the idea that human societies form recurrent social formations by replicating previous studies with different methods and data. Our results also contribute nuance to previously established measures of social complexity, illustrate diverse trajectories of change, and shed further light on the finite bounds of human social diversity.

## Introduction

The emerging model of "recurrent social formations" postulates that only a small number of stable social-ecological configurations exist for human societies [1, 2]. The basic idea is this: one can observe a small set of the same empirical regularities in societies cross-culturally and independent of geography or time. These regularities reflect social and environmental conditions and how these conditions have interacted to settle into an overall configuration.

The model of recurrent social transformations combines qualitative insights from modern bio-economic theory with quantitative insights gained using data reduction and algorithmic techniques from computational science. This empirical approach attempts to avoid the pitfalls of the widely-critiqued idea that societies evolve through a linear series of progressively more complex forms toward an ethnocentrically defined endpoint [1–4]. For example, such

**Data Availability Statement:** All relevant data are within the paper and its Supporting Information files.

**Funding:** The authors received no specific funding for this work.

**Competing interests:** The authors have declared that no competing interests exist.

computational analyses on datasets encoding information on social formations have proven fruitful in the study of social complexity [1, 5, 6]. As these kinds of datasets and analyses continue to emerge, the robustness of previous results to changes in data and method should be explored. Recurrent social formations identifiable by only one method may be nothing more than a mirage—a kind of confirmation bias similar to the phenomenon of *p*-hacking (wherein a single analytical method is misused to artificially create results that are almost certainly false-positive but construed via metrics such as *p*-values to be "significant"). This article contributes to assessing the robustness of recurrent social formations to changes in computational methods and data sets. That is, we use a multi-dimensional clustering algorithm to explore "clumps" in data on human societies indicative of recurrent social formations, and we then compare our results with those identified by researchers using alternative methods and datasets.

In the remainder of this paper, we provide background on the model of recurrent social formations, use clustering to reveal and explore statistically significant typologies of past societies, and we discuss our findings in the context of the conceptual model of recurrent social formations. Our results indicate that the Seshat dataset is robust in that it produces similar results using different methods of analysis. However, our analysis also illustrates the potential to build upon previously-developed methods. In particular, Turchin and colleagues [5] found that the first principal component of the Seshat dataset (PC1) can serve as a useful time-resolved approximation for a society's overall "social complexity." Our results demonstrate that the PC1 metric does not necessarily capture nuance in the diversity of how societies are "socially complex." Indeed, when comparing across regions, there is significant overlap in PC1 between societies of different clusters. In some extreme cases, societies in entirely different superclusters can have similar social complexity factor scores. This nuance provided by a combination of PC1 and cluster trajectories may be of importance for certain research questions. Further, we find that plotting societies along axes of social "scale" and "non-scale" is a robust process that reproduces the same results as a 2018 study by Peregrine [1], who similarly conducted a social complexity cluster analysis on a sample of past societies but using different methods and source data. In the end, multiple methods can provide an important check against confirmation bias and open-up a broader range of research questions for comparative social scientists.

## Background

Comparative social science has a long history of creating typologies of human societies [7]. In anthropology, 19th century theorists proposed that the diversity of human societies resulted from a non-Darwinian evolutionary process in which societies evolve more "Culture" over time at different rates, and Anglo-American societies were viewed as the apex of this universal shared Culture (e.g., [8]). The clearly ethnocentric typologies that resulted from this theory discredited evolutionary anthropology for nearly a half century. However, in the second third of the 20th century, anthropologists again began to study cultural evolution and created typologies of human societies based on empirical characteristics such as family size, levels of political decision making, and subsistence technology (e.g., [7, 9–12]). Although many of these efforts have been criticised as functionalist and overly simplistic, many argue that empirical "social complexity typologies" have a place in modern scholarship [13, 14]. The idea is that comparative social sciences must transition from typologies that are abstract and simple conceptual aids (e.g., band, tribe, chiefdom) to a rigorous, empirical phylogeny of cultures and the kinds of forms they tend to take—similar to biology's transition from pre-Linnean attempts at organism classification to what is now modern biological taxonomy [13, 15].

The emerging conceptual model of "recurrent social formations" attempts to build upon past attempts to categorize human societies in two ways. First, the model reconsiders the

theoretical foundation of empirical typologies by using complex systems theory as a conceptual foundation. That is, the model sees typologies as a consequence of social-ecological interactions rather than discrete "stages" of evolution. For example, complex phenomena (such as predator-prey ecologies, gene regulatory networks, weather systems, etc.) are often mathematically modeled as dynamical systems. These systems are usually discussed with emphasis on the systems' attractors—numerical values towards which the systems tend to evolve. Using the concepts of attractor and of repellor as metaphors, Ullah et. al. [2] use this framework to develop a cluster analysis of the *Standard Cross-Cultural Sample* dataset. They observe four distinct clusters of societies based on features of subsistence, mobility, and demographic variables which, by analogy, may form attractors in the underlying dynamical system governing subsistence behavior.

Similarly, Peregrine conducts an exploratory study that conceptualizes variation in human societies as reflective of adaptive landscapes [1]. Adaptive landscapes describe peaks where the fitness of some combination of traits is high and valleys where fitness is low (e.g., due to the interaction of organisms and their environment). In an analysis of the *Atlas of Cultural Evolution*, Peregrine plots morphological traits of human societies in terms of a "Technology Factor" and a "Scale Factor." The Technology Factor is a composite of variables concerning writing, land transport, social stratification, political integration, technological specialization, and money; the Scale Factor is a composite of variables concerning fixity of residence, agriculture, population density, and urbanization. This study, too, finds two superclusters and several smaller, more refined clusters of societies in this space that, by analogy, may reflect peaks and valleys in the adaptive landscape which further correspond with attractors and repellers in the underlying dynamical system.

Second, researchers have begun to use computational methods to analyse human societies and identify not only clusters of societies with related attributes but also quantify trajectories of change in the social complexity of human societies. Rather than assuming evolutionary stages, this approach attempts to quantify measures of social complexity using an explicit methodology. For instance, Turchin et. al. [5] conduct a principle components analysis of social attributes often considered indicators of social complexity. Consistent with earlier work (e.g., a 1962 scale analysis by Carneiro [16]), they find that these attributes all correlate and that one dimension accounts for a significant amount of variation in social complexity traits (the first principle component of a principle components analysis–PC1 or the social complexity factor). In essence, this PC1 metric creates a reasonable way to quantitatively measure and compare the overall social complexity of societies in different world regions. Although their study only performs a cursory cluster identification of societies that share similar attributes, the study does quantify trajectories of change in social complexity and also suggests that such trajectories share many recurring features cross-culturally.

The above studies suggest that human societies tend to evolve toward a finite set of recurrent social formations; however, this possibility needs further exploration. Recurrent social formations identifiable by only one method or in one dataset may be nothing more than confirmation bias. In the remainder of this article, we ask: Do we find recurrent social formations in the Seshat database that also replicate the trajectories of change in social complexity identified by Turchin and colleagues? Specifically, our study directly builds on the studies above by using a novel clustering algorithm to evaluate how robust the observation of super-clusters and recurrent changes in social complexity are to a change in method and dataset.

First, we use the clustering algorithm in an attempt to replicate Turchin and Colleagues' results. This assesses whether their results are robust to a change in method of data reduction. Second, we attempt to replicate Peregrine's results of two superclusters and several smaller, minor clusters of societies using the Seshat Database. This assesses whether the observation of

two superclusters is robust to changing datasets. In the end, our analysis largely replicates previous findings and adds weight to the emerging model that human societies organize into a finite number of social-ecological configurations constrained by ecology and social evolutionary processes.

## Data and methods

We begin with the *Seshat: Global History Databank* [17]. This dataset encompasses information on over 400 polities from 30 sampling locations across the globe. Seshat encodes social complexity features pertaining to social structures, technologies, information systems, economies, subsistence strategies, and other variables for each polity. This database is designed to measure different aspects of societies and evaluate theories of cultural evolution and the evolution of social complexity [5, 6, 18]. Central to evaluating theories for the evolution of social complexity is a baseline description of differences in social complexity across cases and over time [5].

In accordance with prior analyses, our cluster analysis is conducted on a subset of 51 variables from Seshat that the original authors of the database have deemed reliably identifiable from the archaeological and historical records [5, 6]. These variables encode information such as overall population, largest-settlement population, territory, hierarchy, and boolean variables indicating the presence of various aspects of writing systems, texts in circulation, monetary systems, public infrastructure, and government extent.

Our analysis is conducted on a derivative version of Seshat we have constructed and named *Shiny Seshat*. This iteration upon the original database improves upon the imputation methods used in previous analyses, primes the data to be more appropriately suited for temporally-resolved, polity-wise analysis, and patches a number of human-error typographic mistakes in the original dataset.

### Imputation of missing values

Incomplete entries in the dataset are filled in using statistical imputation—a robust method for performing analysis on incomplete data [19, 20] that has been previously used and explicated in the context of Seshat data [5]. Data entries containing missing information may be subject to systemic bias that has led to their incompleteness; thus, statistical imputation can help alleviate bias in data as opposed to simple list-wise deletion of incomplete entries [20]. Particularly in the archaeological and historical sciences, certain societies and cultures can tend to receive more scholarly attention than other societies and cultures, and this can manifest in Seshat in the form of incomplete data. Therefore, imputation is an important process to represent the greatest amount of social variation in our analysis. However, we were unable to completely replicate the imputation method used by Turchin and colleagues [5].

Fortunately, we managed to improve upon Turchin and colleagues' results using a new open-source imputation tool for Python known as *datawig*, [21] version 0.1.10. This tool utilizes a deep neural network (DNN) that is especially suited for imputing both numeric and non-numeric data. The imputer's parameters (such as number of hidden layers for a feature, hyperparameter optimization options, or feature encoding options) are customizable, but we found datawig's default, largely automatic determination of these parameters to be quite sufficient for our purposes. We specify only to enable hyperparameter optimization and set the number of training epochs to 1,000.

Central to imputation efforts are the replication of nine "Complexity Characteristics" (CCs) encoding information on polity population (*PolPop*), territory (*PolTerr*), largest-settlement population (*CapPop*), hierarchy (*Hier*), government (*Gov*), infrastructure (*Infra*), writing

(*Writing*), written texts (*Texts*), and forms of money (*Money*), respectively [5, 22]. These CCs are useful in that they can serve as broad measures of complexity within these domains even in the absence of completely encoded data. For example, a *Writing* score is assigned based on the values of the "Mnemonic devices," "Non-written records," "Script," and "Written records" features, but only one of these features need be encoded for a given polity to be assigned a *Writing* score.

For each complexity characteristic, we create "regression terms" to input into the imputer in order to provide additional prediction-improving information during the imputation training process. These terms are nearly identical to those indicated in Turchin's piece on fitting regression models to Seshat ([22] pg. 46). In practice, these terms are simply added as additional feature columns. They are:

$$x_{0,i,t} = \sum_{\tau < t} e^{-(t-\tau-100)/100} Y_{i,t-\tau} \tag{1}$$

$$x_{1,i,t} = \sum_{i \neq j} e^{-\delta_{i,j}/1100} Y_{j,t-1} \tag{2}$$

$$x_{2,i,t} = \sum_{i \neq j} w_{i,j} Y_{j,t-1} \tag{3}$$

where $x_{n,i,t}$ is term $n$ for polity $i$ at time $t$ and $Y_{j,t}$ is the value of complexity characteristic $Y$ for polity $j$ at time $t$. Here, $x_{0,i,t}$ helps encode the history of $Y$ by summing all temporally previous values with an exponential discount that grows greater the older the value is relative to $t$. We use an exponential discount of $e^{-(t-\tau-100)/100}$ as series in Shiny Seshat are sampled at the scale of centuries. This produces a factor of $e^0$ for the most recent previous value, $e^{-1}$ for the second most recent value, $e^{-2}$ for the third, etc.

$x_1, i, t$ helps encode spatial diffusion of $Y$ between polities and includes a similar exponential discount factor; $\delta_{i,j}$ is simply the distance between polities $i$ and $j$ in kilometers, and the 1100 kilometer constant originates from optimization conducted in Turchin's work [22]. Finally, $x_{2,i,t}$ helps encode linguistic distance; we let $w_{i,j} = 1$ if polities $i$ and $j$ share a common language, $w_{i,j} = 0.25$ if they share a common language family, and $w_{i,j} = 0$ otherwise (a value between 1 and 0.25 for common linguistic genus was not included as this is not readily coded in the current public version of Seshat).

In previous analyses [5, 22], the fidelity of imputation prediction has been quantified using the $\rho^2$ metric [23]:

$$\rho^2 = 1 - \frac{\sum_i (Y_i^* - Y_i)^2}{\sum_i (\bar{Y} - Y_i)^2} \tag{4}$$

Where each $Y_i$ are the actual observations in the test set, $\bar{Y}$ is the mean of all $Y_i$, and $Y_i^*$ are the predicted values. Using this function, $\rho^2 = 1$ is a perfect prediction, $\rho^2 = 0$ is a prediction just as good as simply replacing missing values with the mean of known values, and anything less than zero is a worse prediction than simply predicting all values to be the mean of all $Y_i$. However, when working with the Seshat data, we find on some occasions that we encounter the edge-case of $\bar{Y} = Y_i \ \forall \ i$ (when the perfect prediction *is* the mean of the data), leading to a division by zero.

Specifically, this edge-case arises during the imputer's training step when the equation is used to optimize predictive fit on segments of data automated via $k$-fold cross validation. Every so often, the algorithm happens to sample a subset of data from an integer-valued

column (e.g., "Administrative levels") where every data point happens to be the same (e.g., a subsample of polity-centuries that all happen to have three administrative levels). Thus, the perfect prediction (three administrative levels) *is* the mean of the data, and a division by zero occurs and crashes the imputation program.

Thus, we modify the equation and create a new function, $\rho_\mu^2$, to include this additional case:

$$\rho_\mu^2 = \left\{ \sum_i - |Y_i^* - \bar{Y}| \bar{Y} + 1 \text{when } \bar{Y} = Y_i \ \forall \ i \ 1 - \frac{\sum_i (Y_i^* - Y_i)^2}{\sum_i (\bar{Y} - Y_i)^2} \text{otherwise} \right. \tag{5}$$

Should the $\bar{Y} = Y_i \ \forall \ i$ edge-case occur, this places $\rho_\mu^2 = 1$ as a perfect prediction of all values being the mean with $\rho_\mu^2$ increasingly less than one as predicted values diverge from the mean. In this manner, the usefulness of the metric is maintained in all cases, despite $\rho_\mu^2 = 0$ being semantically meaningless in the edge-case.

Once each regression variable is coded for, we begin training and testing the imputer on data with known values. We estimate the fidelity of each CC prediction using 5-fold cross-validation. Table 1 indicates these results. In practice, we find that, for this case, $\rho_\mu^2 \approx \rho^2 \approx R^2$.

During exploratory analysis, we discovered that *not* including spatial and linguistic distance led to better predictions for every CC except for *Texts* and *Money*. We hypothesize this is due to the imputer's DNN being somewhat sensitive to including too much irrelevant information during the training phase. This hints at the possible theoretical consequence that cultural diffusion may, then, be a largely irrelevant factor in the development of many societal characteristics. However, exploring this implication is beyond the scope of this study. Thus, we leave it at that and only include $x_{1,i,t}$ and $x_{2,i,t}$ for the *Money* and *Texts* variables to improve the overall prediction accuracy.

After each CC is imputed, we further impute every missing value in all other columns in Seshat, allowing the imputer to use the already-imputed CCs as input. Additionally, the imputer allows for the possibility of imputing non-numeric values. We utilize this to impute categorical features such as "bureaucracy source of support," "degree of centralization," "linguistic family," etc. We exclude from imputation only features indicating proper names of cultures, places, and rituals. Predictive power for the individual variables is comparable to that of the CCs themselves.

## Data cleaning and reorganization

Beyond imputation of missing values, the most immediately recognizable difference between Seshat and Shiny Seshat is that Shiny Seshat reorganizes information from individual listings

**Table 1. Fidelity metrics for the prediction of each Complexity Characteristic (CC).**

| Complexity Characteristic | $\rho^2 \approx \rho_\mu^2 \approx R^2$ |
|---|---|
| PolPop | 0.86 |
| PolTerr | 0.58 |
| CapPop | 0.80 |
| Hier | 0.89 |
| Gov | 0.92 |
| Infra | 0.88 |
| Writing | 0.88 |
| Texts | 0.95 |
| Money | 0.79 |

of data points into a matrix-like format where rows are polities during a specific century and columns are features. For brevity, we dub each "polity during a specific century" as a "polity-century," indicating that the row is not only sociopolitically distinguished but temporally distinguished as well, e.g. "The Ottoman Empire 1500CE-1600CE" or "Woodland Cahokia 300BCE-200BCE." This avoids previous ambiguity using the term "polity" without regards to the polity's internal chronology, as a single polity's features will usually take on multiple values throughout its tenure. Thus, time-resolved analyses are done at the scale of *polity-centuries* rather than at the scale of *polities*.

The following changes are also made:

- Seshat encodes binary features on a scale of "present," "inferred present," "inferred absent," and "absent." Mirroring previous work [6, 22], these features are converted to the numeric forms of 1.0, 0.9, 0.1, and 0.0, respectively

- Values encoded as ranges in Seshat are stored as medians in Shiny Seshat. For example, if Seshat indicates that a particular polity has between 6 and 7 administrative levels (ranges such as this typically indicate uncertainty and/or organizational complexity), we encode this as "6.5" administrative levels. For analytic purposes, this simplifies the encoding while still representing the full information for nearly all ranges.

- Polity-centuries spanning multiple Natural Geographic Areas (NGAs) are also more clearly indicated as such in Shiny Seshat in the form of a simple list of NGAs for each polity-century. If one wishes to compare NGAs instead of individual polity-century (such as we do for cluster trajectories in the following sections), it requires only a few simple data transformations to wrangle the dataset into an appropriate form.

- We perform a principal component analysis in the same manner as Turchin et. al. [5] and include the first principal component (PC1) for each polity-century, though this component differs slightly from the one from Turchin et. al.; the PC1 of Shiny Seshat only accounts for 68% of the variance (our code which performs this is included in S2 File). PC2 through PC6 account for 13%, 7%, 6%, 4%, and 2% of the remaining variance, respectively, while PC7 through PC9 all account for less than one percent of variance. Another difference from the original dataset is that the eigenvalue for our PC2 exceeds the standard significance threshold of 1.0; see the S1 Appendix for details on the PCA.

## Algorithm

Sparse Subspace Clustering (SSC) is a clustering algorithm capable of efficiently dealing with sparse, highly-dimensional data [24]. The algorithm is resilient to missing, erroneous, and noisy data, and the algorithm is not overly sensitive to data points near subspace intersections. The number of clusters $k$ need not be known prior to clustering; however, a handful of hyper-parameters are still required for the algorithm to function. The algorithm is summarized from the work of Elhamifar and Vidal [24] in Algorithm 1.

**Algorithm 1** Sparse Subspace Clustering (adapted from Elhamifar and Vidal [24])

```
Input: A matrix of data points Y
  1. Solve the sparse optimization program (Eq 8)
  2. Normalize the columns of C as cᵢ ← cᵢ/‖cᵢ‖∞
  3. Form an adjacency matrix W = |C| + |C|ᵀ
  4: Perform spectral clustering on W
Output: Cluster labels for the data points in Y
```

We choose to cluster using SSC as Seshat is, indeed, quite sparse in some categories prior to imputation, highly dimensional, and contains lower-dimensional "subspaces" with meaningful interpretations (our typologies in question).

The algorithm operates on the principle of "self-expressiveness" [24]. That is, we start by assuming that every data point $y_i$ can be expressed as a linear combination of every other point (with a total of $n$ points):

$$y_i = c_{i0}y_0 + c_{i1}y_1 + \cdots + 0 \cdot y_i + \cdots + c_{in}y_n \tag{6}$$

In essence, a greater weight $c_{ij}$ indicates that data point $y_j$ belongs in the same cluster as $y_i$, and a weight $c_{ij}$ approaching 0 indicates that $y_j$ is in a different cluster from $y_i$.

In the semantics of polities as data points, this means the algorithm operates on the assumption that no human society has a single feature of social complexity that is entirely unique. That is, a polity's quantitative particularities can always be expressed as some weighted mixture of the aspects of other polities.

Now, we define a matrix $y = [y_1 \cdots y_n]$ and formulate the equation

$$Y = YC + E + Z, \quad \text{diag}(C) = 0 \tag{7}$$

where $C$ is a matrix of weights, $E$ is a matrix to account for error in the dataset, and $Z$ is a matrix to account for noise in the dataset [24]. We wish to find a $C$, $E$, and $Z$ that solve Eq 7, thus we frame this as a sparse optimization program

$$\min \quad \| C \|_1 + \lambda_e \| E \|_1 + \frac{\lambda_z}{2} \| Z \|_F^2 \tag{8}$$
$$\text{such that } Y = YC + E + Z, \quad \text{diag}(C) = 0$$

where $\lambda_e = \alpha_e / \sqrt{n}$, $\lambda_z = \alpha_z / \sqrt{n}$, and $F$ indicates the Frobenius norm [24]. We utilize an Alternating Direction Method of Multipliers (ADMM) optimizer (algorithm also provided by Elhamifar and Vidal [24]) to solve this program.

Lastly, we formulate a matrix $W = |C| + |C|^T$. This matrix serves as an adjacency matrix for a graph—effectively turning linear combination weights between data points into edge weights between nodes. Using a hyperparameterized threshold $\rho$ to determine when a weight is too small to indicate a connection, we are left with a graph containing a discrete number of connected components. These components encode the clustering [24]. In practice, we simply count the number of connected components and feed the graph into a spectral clustering function [25] to create labels for the data points in Y.

Our implementation of this algorithm is available as Python code (S1 File).

## Cluster optimization

We begin by sampling from Shiny Seshat the 51 variables of analysis used in prior works (see Whitehouse et. al. [6] Extended Data Table 5 for a full list of these variables); this is the subset of data that we will perform clustering on. We then normalize each of these features using min/max normalization. SSC involves a high number of matrix multiplications, so this prevents floating-point overflow while still maintaining sufficient information to perform clustering. We also collapse polity-centuries into data points representing the entire base polity by simply taking the mean across all time periods. Further, we found that including too many highly-imputed data points diminished the algorithm's ability to converge on a good clustering. Thus, for the analysis, we have paired down the dataset to only include data points with at least 75% encoding for the fifty-one features, leaving us with 271 polities.

Our primary means of gauging how good a clustering we have is silhouette analysis [26]. A "silhouette coefficient" between -1 and 1 is calculated for each data point. This coefficient is a measure of how close a data point is to the center of the cluster it has been assigned. A silhouette coefficient close to 1 indicates that a data point is very near the center of its assigned cluster. Conversely, a silhouette coefficient close to -1 indicates that a data point is much closer to a *different* cluster's center than it is its own cluster's center. A silhouette coefficient close to 0 indicates that a data point is near a boundary between clusters roughly equidistant between the centers of its assigned cluster and another cluster.

SSC requires a number of hyperparameters. We use hyperopt, a hyperparameter optimization library [27], to select hyperparameters and find a clustering that minimizes the number of data points with a negative silhouette coefficient. We found, however, that this process does not converge upon the most optimal labelling but rather a labelling that is "fairly close" to optimal. Thus, after performing automatic clustering, we manually optimize the labeling by iterating over data points with a negative silhouette to relocate them to the cluster where they have the highest silhouette. Specially, we simply iterate through each negative-silhouette data point, re-compute is hypothetical silhouette coefficients were it belonging each of the clusters, and re-label it to whichever cluster in which it has the highest silhouette coefficient. Alternatively, using other optimization criteria that do not involve the silhouette coefficient such as MDL or information entropy would perhaps provide better, more streamlined automatic clustering should this process need to be carried out again in future analyses.

Fig 1 provides a silhouette plot for each data point. We indicate the most "archetypal" polities for each cluster in Table 2. These are the polities with the highest silhouette coefficient in their given cluster. The average silhouette score across all clusters is 0.18. Although no standard exists for what constitutes a "significant" average silhouette (especially in the social sciences), we can compare this score against a score distribution obtained from performing the same clustering process on similar data sets constructed randomly.

We construct a dataset in the same shape and general form as our clustering input dataset, but instead filled with uniformly random values. It contains the same number of rows and columns corresponding to each entry of actual data. For each column, we note the minimum and maximum values taken on by the actual data, and generate a new uniformly random number within these bounds for each entry. We then perform optimized clustering on this dataset and calculate its silhouette score.

We repeat the above process 540 times and collect the results. We find that the mean silhouette score from clustering each of 540 random data sets is −0.15 and that the silhouette score exceeds 0.18 in only 3.9% of cases. In other words, there is a low probability that we are detecting a "false positive" cluster signal in the Shiny Seshat data. In terms of qualitative significance of the clustering, we explicate the uniqueness and significance of individual feature distributions in the following section.

## Results

To sum up our results, we conceptually replicate previous studies. We find two equally viable clusterings with insignificantly different average silhouette coefficients: A five-cluster solution, and a two-supercluster solution. In this supercluster solution, Clusters 0 and 1 from the five-cluster solution are grouped into a *Supercluster A* and Clusters 3 and 4 form *Supercluster B*, with Cluster 2 from the five-cluster solution split arbitrarily between the two superclusters. The five-cluster solution clearly covaries with Turchin and colleagues' PC1 social complexity factor. Similarly, we replicate Peregrine's two supercluster morphospaces. The observation of recurrent social formations is robust to our new method and change in dataset. We also

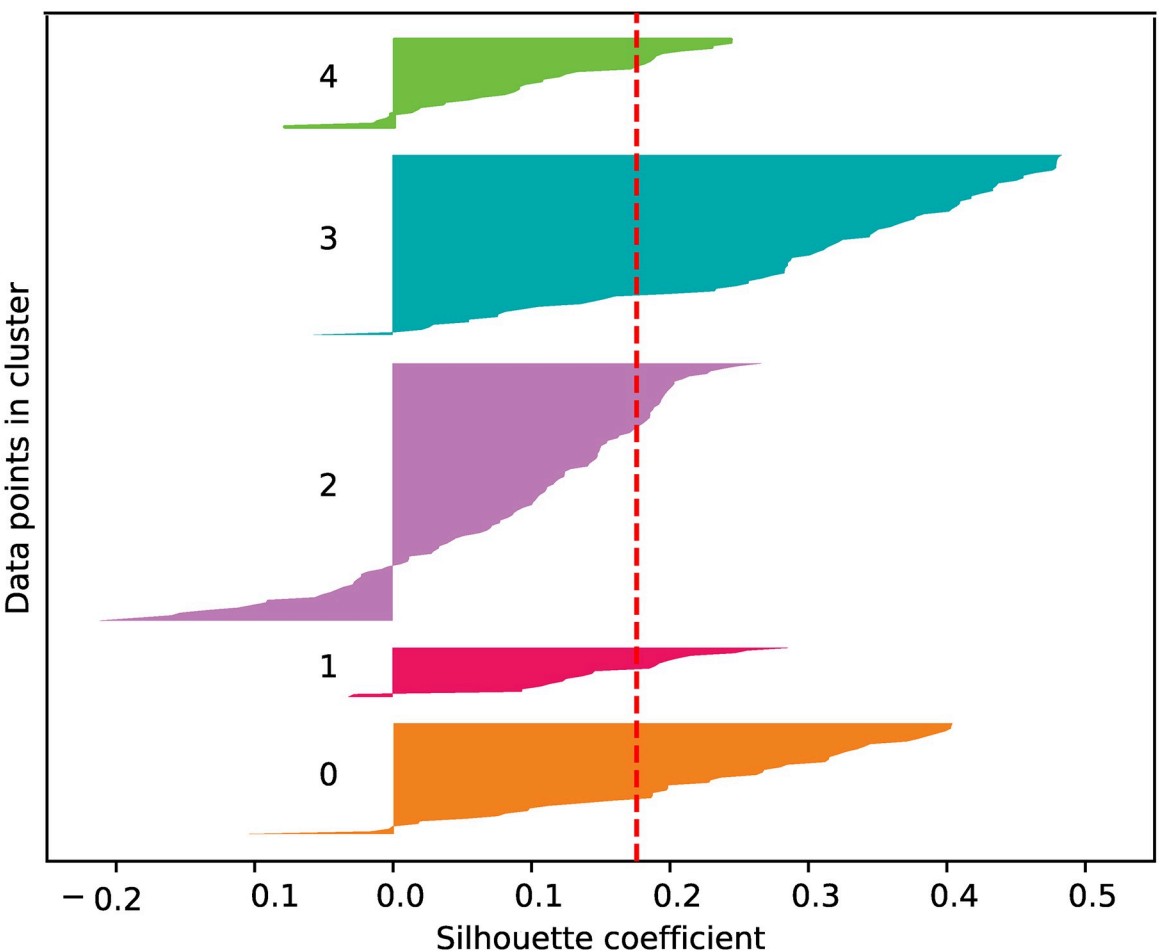

**Fig 1. Silhouette plot for each data point categorizedinto clusters.** The dashed line indicates the average silhouette of 0.18.

document that "cluster trajectories" tell regional histories of how societies evolve and move between clusters in the longue durée. These patterns provide a foundation for understanding the causal forces that drive changes between forms of society.

## Cluster analysis vs. the social complexity factor

Using the first principal component (PC1) of the Seshat dataset (which both our analysis and that of Turchin et. al. [5] found to encode a majority of the data's variance), we may quantify each polity's complexity in terms of the variables encoded in Seshat. The PC1 metric

**Table 2. Archetypal polities.** These are the polities which have some of the highest silhouette scores in their respective clusters.

| Cluster # | Archetypal Polity | Approx. Era | Mean PC1 |
|---|---|---|---|
| 0 | Woodland Cahokia | 600 BCE–700 CE | -2.6 |
| 1 | Cahokia Proper | 1100 CE–1300 CE | -1.5 |
| 2 | Roman Kingdom | 700 BCE–500 BCE CE | 0.8 |
| 3 | Papal States | 1500 CE–1600 CE | 2.6 |
| 4 | Ottoman Empire | 1600 CE–1900 CE | 3.0 |

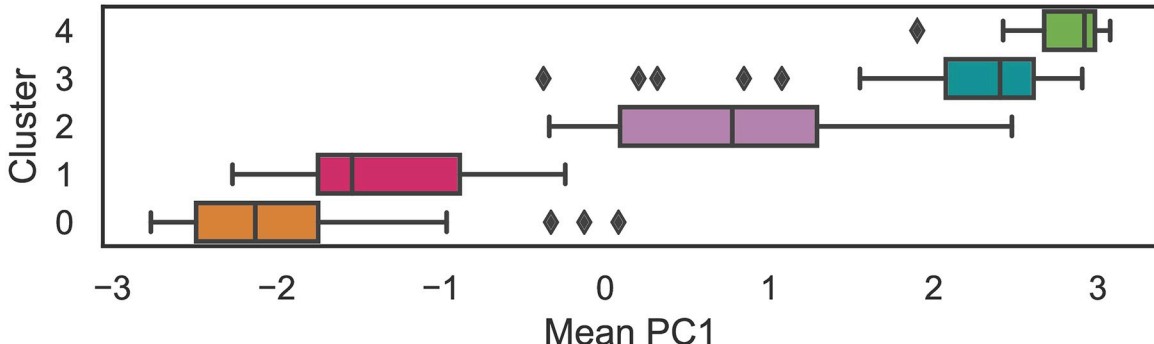

**Fig 2. Distribution of Social Complexity (PC₁) values for clustered polities.**

exemplifies the organization of the five clusters into two superclusters, with Clusters 0 and 1 having generally low PC1 values, Clusters 3 and 4 having generally high PC1, and Cluster 2 having a large variance in PC1 "bridging the gap" between the two superclusters (Fig 2). This trend extends to many of the distributions of the clusters' complexity characteristics; we generally see a "clumping" of superclusters with Cluster 2 spanning a large variance in the middle, or a near-linear scaling of distribution means (S1 Fig).

We also analyze the distributions of Complexity Characteristics (CCs) between clusters (S1 Fig). We see that, for example, social hierarchies within clusters are roughly normally distributed and that more socially complex clusters tend to have taller hierarchies (Cluster 0 has shorter hierarchies than Cluster 1, Cluster 1 has shorter hierarchies than Cluster 2, etc). Mann-Whitney U tests indicate that nearly all distributions of Complexity Characteristics (CCs) between clusters are unique ($p < 0.05$ in all cases and $p < 0.001$ in nearly all cases). What is more interesting for revealing the nature of these clusters is seeing which distributions are *not* likely to be unique; namely, *CapPop* for Clusters 1 and 2 ($p > 0.26$), *Hier* for Clusters 1 and 2 ($p > 0.11$), *Money* for Clusters 0 and 1 ($p > 0.12$), *Money* for Clusters 2 and 3 ($p > 0.21$), *Writing* for Clusters 0 and 1 ($p > 0.35$), and *Writing* for Clusters 3 and 4 ($p > 0.26$).

These results indicate two things. First, there is clear reason to differentiate subclusters within their superclusters, but the *Money* variable may be measuring something that unifies clusters into superclusters in the first place. Second, it seems the development of a written script is almost synonymous with a society existing in any of Clusters 2, 3, or 4. Whether this relationship is causal or simply highly correlated is yet to be explored.

Table 2 lists some of the most archetypal polities in each of our four clusters. These polities are considered among the best fit data points in their respective clusters, and are thus especially representative of the typical quantitative characteristics of the polities in each cluster. This allows us to discuss cluster characteristics in terms of actual historical examples. The Seshat Knowledge Graph [17] provides qualitative info on these polities.

Exemplary of Cluster 0, Woodland Cahokia is the period prior the rise of the urban city of Cahokia proper. Populations were small and foraging was important for subsistence in this period. Cultures in this period practiced mound-building and pottery, and there is evidence for some high-status burials and crop cultivation in the latter half of the period. Cahokia proper is exemplary of Cluster 1, with the sudden emergence of Cahokia as a immense and densely populated center with a population capable of great feats of cooperation such as mound-building and constructing large wooden palisades.

The Roman Kingdom period stands out as a Cluster 2 society as the small, disjoint villages of the Copper, Bronze, and Iron age (Cluster 0 societies) give way to the beginnings of Rome

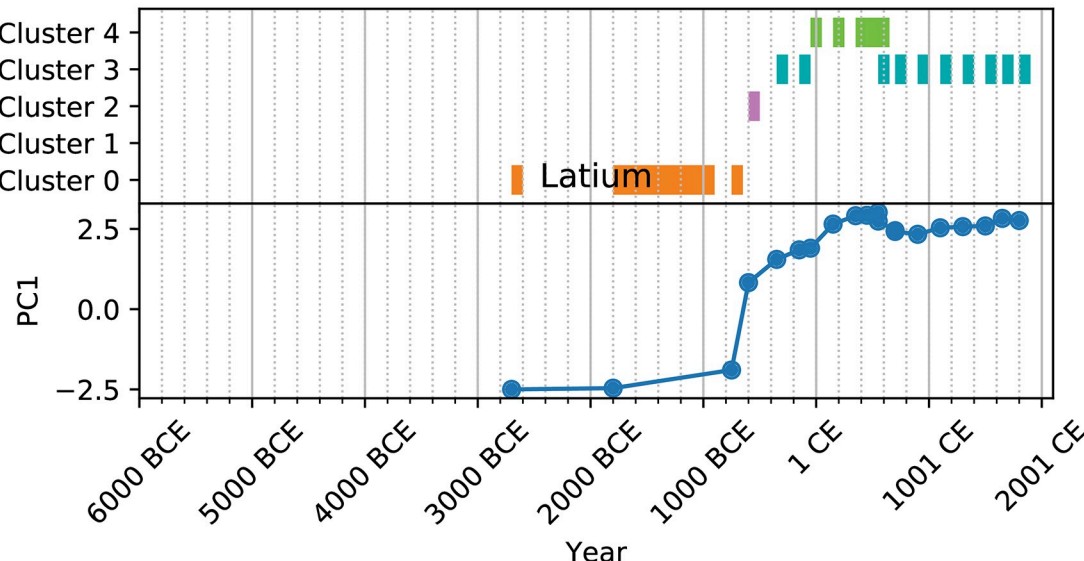

**Fig 3. Cluster trajectory for Latium.**

as a city-state and, following the Kingdom period's conclusion, to the rise of the Roman Republic (a Cluster 3 society). Despite Rome's classification as a Cluster 4 society during the Principate and the Dominate, we see a return to Cluster 3 following the Empire's collapse and a thorough settling-in to this cluster as the Papal States become quantitatively exemplary of Cluster 3 (see Fig 3 to follow this journey).

The Ottoman Empire stands out with the highest silhouette score in Cluster 4. The vast territory of the empire stands in contrast to the relatively small region of the Italian peninsula encompassed by the Papal States. Although the Ottoman empire had a shorter religious hierarchy in comparison to the Papal States, political hierarchies are matched or greater. Further, the Ottoman Empire was consistently a unitary state throughout its tenure, whereas the Papal States' degree of centralization fluctuated over the centuries from strong singular bureaucracies to loose associations of cities. Yet, both of these Supercluster B societies represent a much greater state of social complexity than societies of Supercluster A.

Further, the clustering allows us to create a kind of analog for the social complexity trajectories of natural geographic areas (NGAs) provided by the PC1 metric (Figs 3, 4, 5 and 6). These cluster trajectories illustrate a similar journey through time of cluster membership for the polities occupying an NGA. Temporally, NGAs almost always begin in Cluster 0 and eventually move on to the other clusters. Long-term shifts in cluster membership are usually accompanied by large shifts in PC1 (such as with Fig 3), whereas more rapid fluctuations between clusters usually are accompanied by a relatively stable, if noisy, PC1 (such as with Fig 5). Notably, societies *never* remain in Cluster 2 for the amount of time recorded for the other clusters, suggesting that societies in Cluster 2 are perhaps in an unstable or transitional state. In all cases, time spent in Cluster 2 is typically limited to 200-500 years, whereas time spent in all other clusters can stretch on for millennia (Fig 7a). Further, when accounting for all trajectories, we observe cluster shift frequencies that indicate the vast majority of societies leave Supercluster A without returning, societies tend to pass through Cluster 2 on to Supercluster B, and a majority of societies do not leave Supercluster B once they have entered, and those that do are likely to return (Fig 7b).

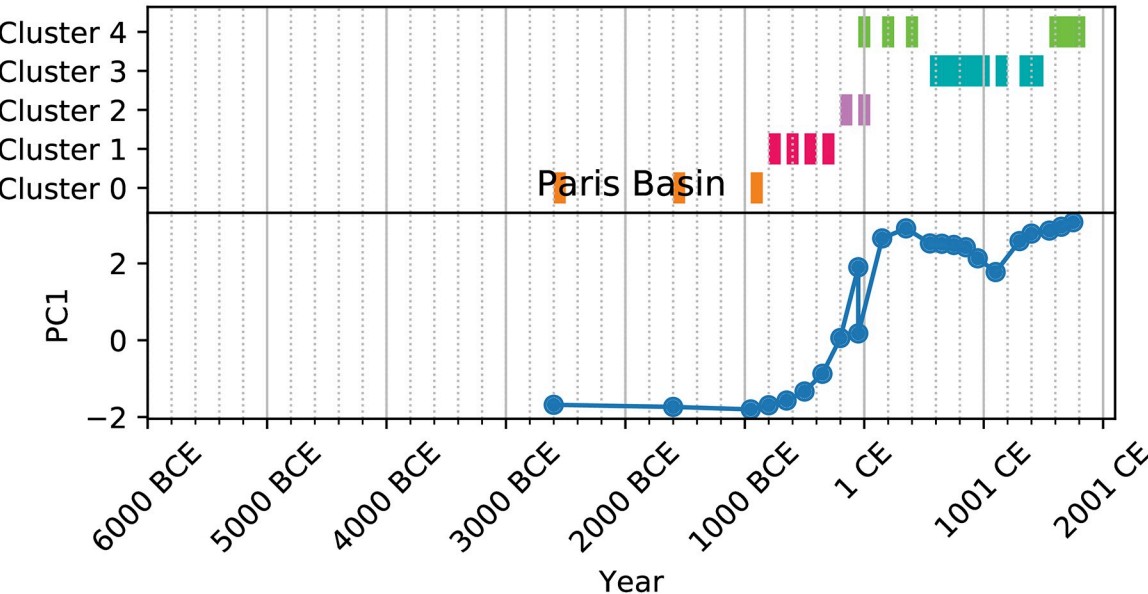

**Fig 4. Cluster trajectory for the Paris Basin.**

Not all geographic areas are very complete in their cluster trajectories due to their data sparseness. The trajectories presented here have been chosen as they are among the most complete trajectories and display dynamics exemplary of their siblings (see the S3 File for all generated cluster trajectories).

## The social complexity morphospace

Building from our cluster analysis, the data replicate the observation of an empirical morphospace of societal scale and technology as observed in a 2018 study by Peregrine [1]. Peregrine's

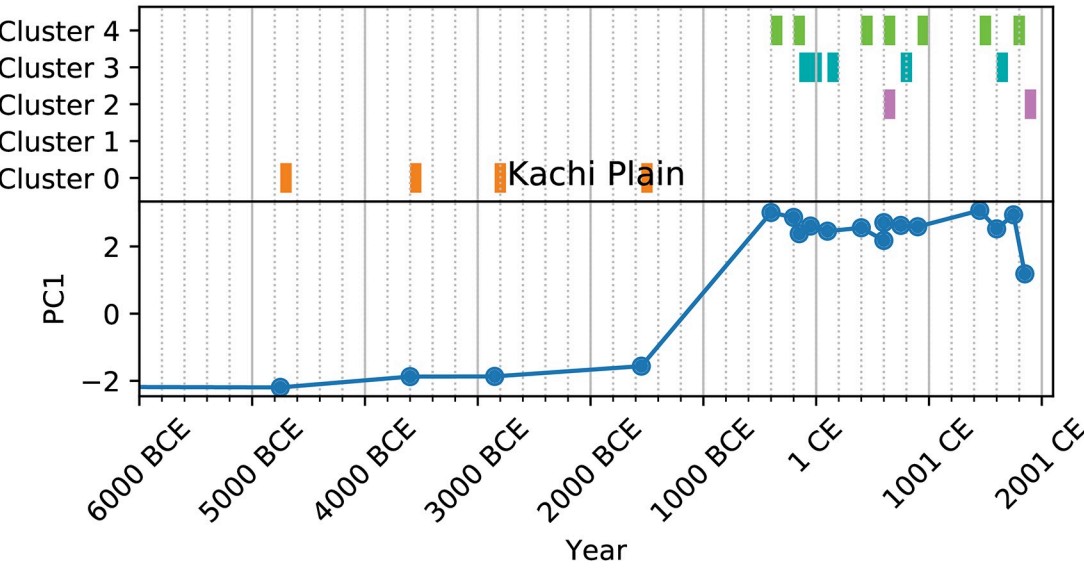

**Fig 5. Cluster trajectory for Kachi Plain.**

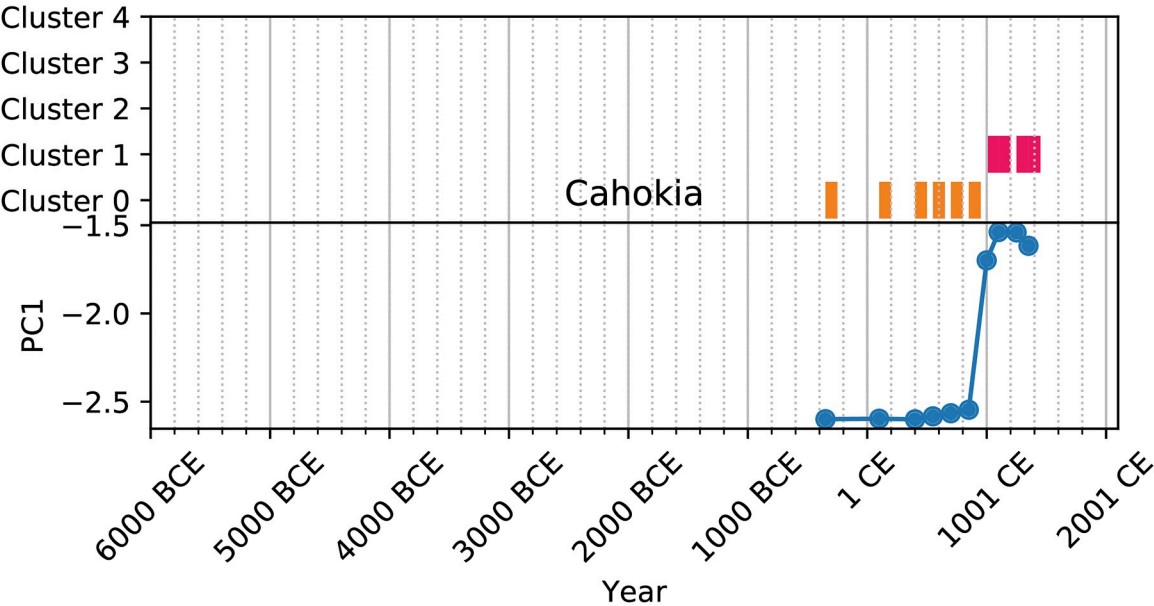

**Fig 6. Cluster trajectory for Cahokia.**

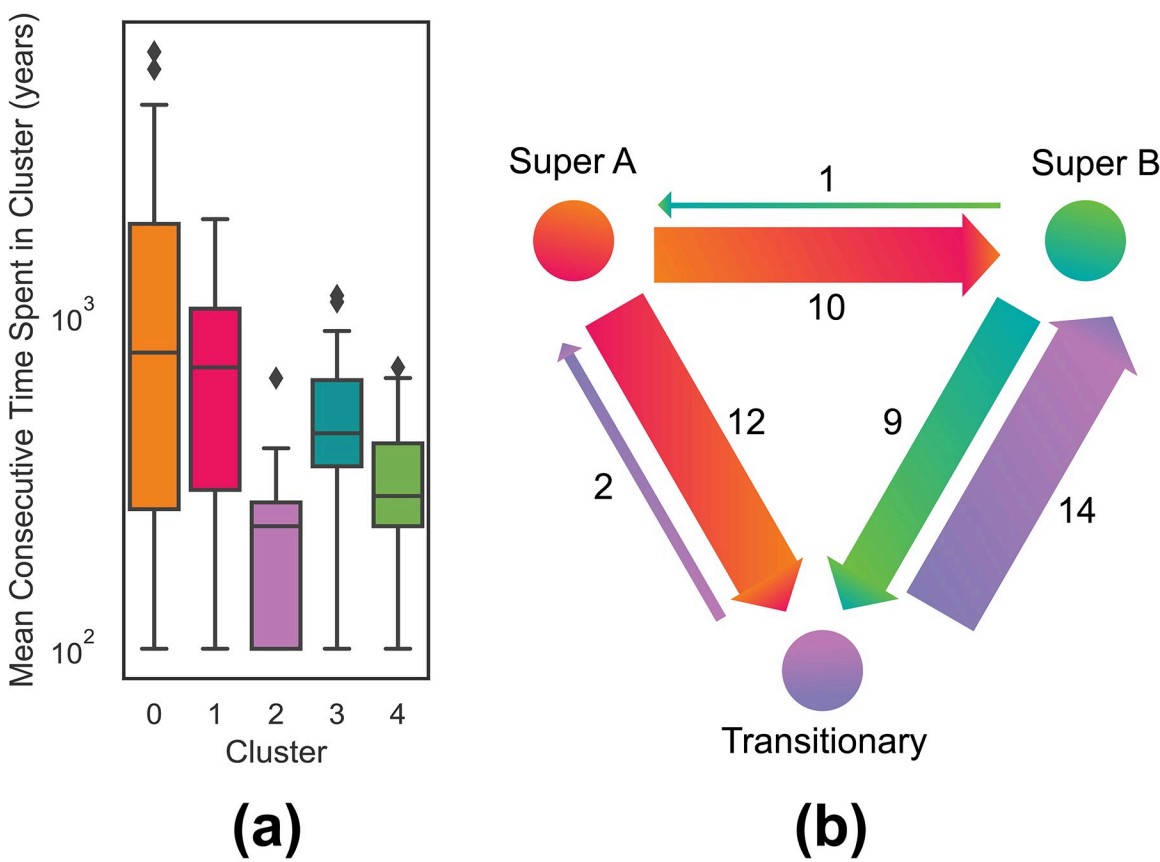

**Fig 7.** (**a**). The mean consecutive time societies spend in each cluster. (**b**) Diagram of the total number of observed societal transitions between Supercluster A, the transitionary Cluster 2, and Supercluster B. Arrow width is proportional to each count.

study analyzes a different dataset, the *Atlas of Cultural evolution*, and uses a different cluster-revealing methodology involving Guttman scaling and morphospace analysis. The *Atlas* encodes similar information to Seshat; to help reduce the data's dimensionality, Peregrine utilizes scale and technology factors derived from the Murdock-Provost scale of cultural complexity [28] by Chick [29]. From Peregrine's data, the Technology Factor is a composite of variables concerning writing, land transport, social stratification, political integration, technological specialization, and money; and the Scale Factor is a composite of variables concerning fixity of residence, agriculture, population density, and urbanization.

We present simple, roughly analogous alternatives to Peregrine's Scale and Technology factors derived from Seshat data alternatively named factors of "Scale" and "Non-scale" for greater clarity and rigor. We categorize Seshat's population, territory, and hierarchy features as features of scale and all other features (such as variables concerning infrastructure, writing, and economy) as features of non-scale. In the same manner as we did prior to clustering, we then min/max normalize all features of scale to be between 0 and 1. We then create a scale factor for each polity by simply summing together each polity's normalized scale features. Since all non-scale features are binary in nature (Shiny Seshat encodes them as present with a 1 and absent with a 0, with varying degrees of uncertainty assigned intermediary values), we finally assign polities a non-scale factor that is analogous to the total number of non-scale features that are listed present for each polity. This method is further replicated by creating axes of normalized Scale CCs (*PolPop*, *PolTerr*, *CapPop*, and *Hier*) and Non-scale CCs (*Govt*, *Infra*, *Writing*, *Texts*, and *Money*).

In Fig 8, we plot polities along the axes of scale and non-scale factors. This plot shows almost precisely the same morphospace curve as Peregrine's study [1], and the plot also shows that our own clusters are quite clearly clumped together in this space. Density analysis of the space exemplifies the two superclusters that polities tend to exist in, as well as an additional, smaller smattering between the two clusters representing the centroid of Cluster 2. These results are consistent with the proposition that human societies are well described by recurrent social formations driven by underlying social-ecological interactions.

## Discussion & conclusion

In this analysis, we have algorithmically uncovered discrete clusters of societies based on features of government, economy, technology, religion, military, information systems, and population variables provided by the *Seshat: Global History Databank*. Analysis indicates that solutions of two and five clusters are the best fit to Seshat's data, with lower-complexity Clusters 0 and 1 in the first supercluster, higher-complexity Clusters 3 and 4 in the second supercluster. Cluster 2 is a kind of intermediary, "transient" cluster of societies transferring between the two superclusters. Results hint at the possibility of the development of a written script playing a role in the shift from the first supercluster to the second, although further exploration is needed to determine if this relationship is causal or simply highly correlated.

Our cluster trajectories indicate that, while Seshat and the corresponding Social Complexity (PC1) metric are resilient to differing methodologies, the PC1 metric does not always capture much of the diversity between societies. Indeed, two societies with different technologies, social organizations, and cluster membership may be calculated to have a near-identical PC1. Tying in cluster analysis to study societies in terms of both PC1 and typology may be of use to scholars seeking to utilize a more comprehensive approach to quantification. For example, large changes in PC1 were shown to cross-culturally precede the development of judgemental deities [6], thus contradicting the Moralizing Gods hypothesis. We hypothesize that these large changes in PC1 may also temporally coincide with shifts in cluster membership.

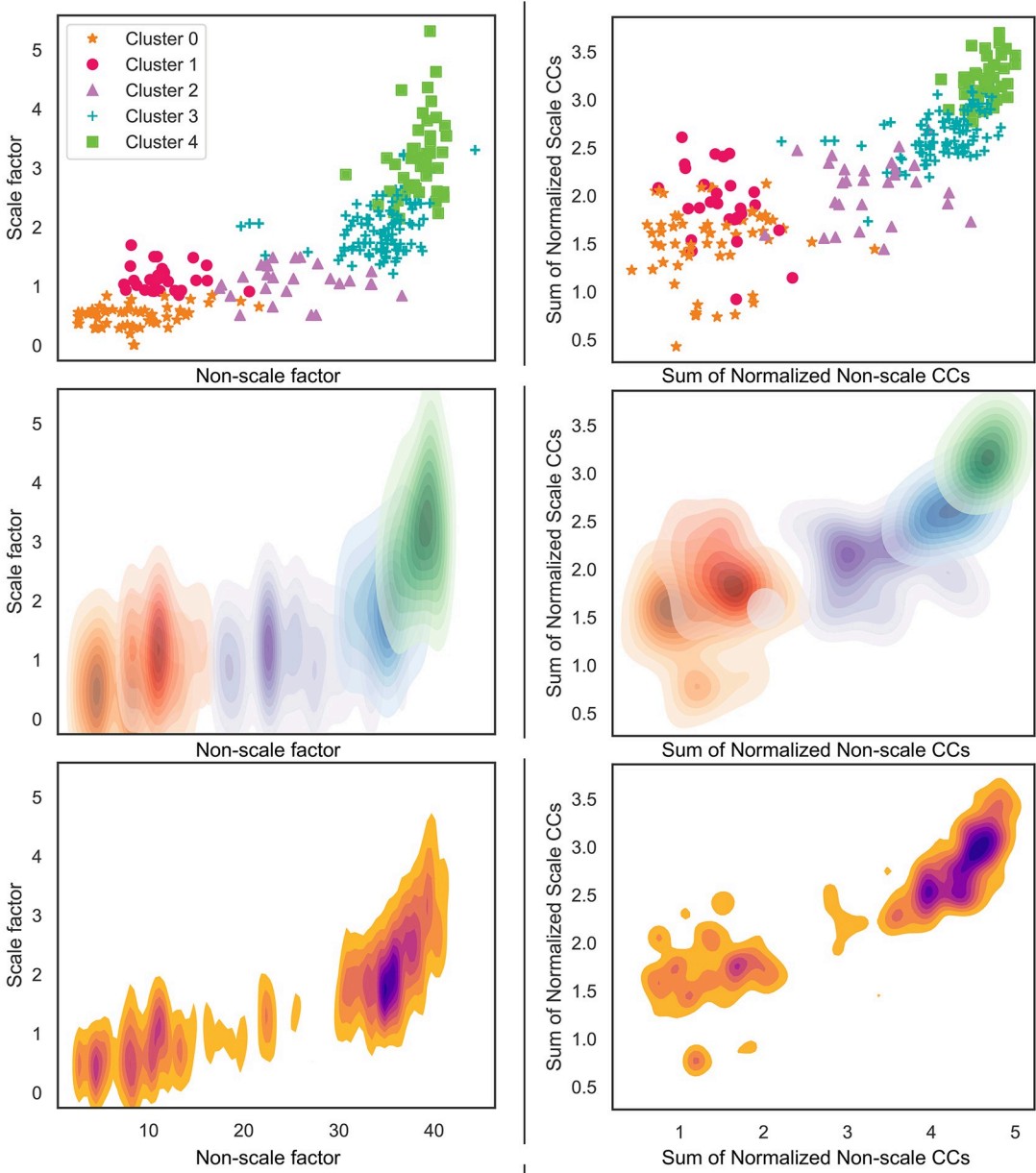

**Fig 8. The empirical morphospace of cultural complexity: All datapoints by cluster (top plots), per-cluster probability density (middle plots), and overall probability density (bottom plots).**

We offer the interpretation of our results within the framework of dynamical systems theory. We hypothesize that there exists an underlying model with attractors in the space of scale and non-scale that manifests in the form of the clusters that we have found. With this interpretation, our analysis is ultimately more exploratory than explanatory. Though, our results offer robustness to the theory behind recurrent social formations. Our methods and dataset differ wholly from those of the study by Peregrine [1], yet we seem to observe the very same phenomena of social complexity morphospaces. In the future, predictive mathematical models should be constructed to describe the attractors that lead to clumping in the morphospaces and shed light onto the dynamics that create these apparent "social steady-states".

## Supporting information

**S1 Fig. Distributions of Complexity Characteristics (CCs) across clusters.** Using a kernel-density estimation.
(TIF)

**S2 Fig. PC2 plotted against PC1with datapoints by cluster (top), per-cluster probability density (middle), and overall probability density (bottom).**
(TIF)

**S1 File. Clustering and plot generation code.** We include our Python 3 implementation of the clustering algorithm and all analysis and plot-generation code.
(ZIP)

**S2 File. Shiny Seshat scrubbing code.** We include our Python 3 program that begins with the original, untampered Seshat database and performs the entire process of turning it into Shiny Seshat (including all error correction, Complexity Characteristic creation, imputation of missing values, etc.).
(ZIP)

**S3 File. A complete collection of cluster trajectories.** We include trajectories for all Natural Geographic Areas (NGAs) for which there is sufficient data (all polities with at least 75% complete encoding for the 51 features of analysis; see the Data and methods section for details).
(ZIP)

**S1 Appendix.**
(PDF)

## Acknowledgments

The authors would like to thank their colleagues for feedback and proofreading as well as their spouses for emotional support.

## Author Contributions

**Conceptualization:** Lux Miranda.

**Data curation:** Lux Miranda.

**Formal analysis:** Lux Miranda.

**Investigation:** Lux Miranda.

**Methodology:** Lux Miranda, Jacob Freeman.

**Project administration:** Lux Miranda, Jacob Freeman.

**Resources:** Lux Miranda.

**Software:** Lux Miranda.

**Supervision:** Jacob Freeman.

**Validation:** Lux Miranda.

**Visualization:** Lux Miranda.

**Writing – original draft:** Lux Miranda.

**Writing – review & editing:** Lux Miranda, Jacob Freeman.

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
