## [Decision Letter · Decision Letter 0]

24 Dec 2019

PONE-D-19-30886

The two types of society: mathematically revealing recurrent

social formations and their evolutionary trajectories

PLOS ONE

Dear Mx. Miranda,

Thank you for submitting your manuscript to PLOS ONE. After careful consideration, we feel that it has merit but does not fully meet PLOS ONE’s publication criteria as it currently stands. Therefore, we invite you to submit a revised version of the manuscript that addresses the points raised during the review process.

While Reviewer 2 raised some minor comments, Reviewer 1 showed major concerns.

Reviewer 1 actually responded No to a template question: "Has the statistical analysis been performed appropriately and rigorously?"

One of the main goals of this study is applying a new statistical analysis. 

The authors confirmed the previously-found claim (PC1 explains large part of cultural variations in historical societies, and thus societies can be classified by a single measure) by using another method.

Thus, it is very important to perform appropriate and rigorous statistical analysis and also describe their method in detail.

Details of a deep neural network (DNN) should be provided, otherwise nobody can reproduce the results.

I also agree with Reviewer 1 that the authors should show some statistical results and figures about PC2, which will clarify how PC1 is important (or not). Since PC1 accounts for 61% of the variance, cluster analysis and PCA results must be similar to some extent. Reviewer 1 found this point 'tautological.' 

Additional statistical results comparing their clustering analysis and PC2 (or, PC2 and higher) should be provided.

All comments by Reviewer 2 are also quite reasonable.

Please adequately respond to the comments by both reviewers and revise the ms accordingly.

We would appreciate receiving your revised manuscript by Feb 07 2020 11:59PM. To enhance the reproducibility of your results, we recommend that if applicable you deposit your laboratory protocols in protocols.io, where a protocol can be assigned its own identifier (DOI) such that it can be cited independently in the future. For instructions see: http://journals.plos.org/plosone/s/submission-guidelines#loc-laboratory-protocols

We look forward to receiving your revised manuscript.

Kind regards,

Joe Yuichiro Wakano

Academic Editor

PLOS ONE

Journal Requirements:

Reviewers' comments:

Reviewer's Responses to Questions

**Comments to the Author**

1. Is the manuscript technically sound, and do the data support the conclusions?

Reviewer #1: Partly

Reviewer #2: Yes

2. Has the statistical analysis been performed appropriately and rigorously? 

Reviewer #1: No

Reviewer #2: Yes

3. Have the authors made all data underlying the findings in their manuscript fully available?

Reviewer #1: Yes

Reviewer #2: Yes

4. Is the manuscript presented in an intelligible fashion and written in standard English?

Reviewer #1: Yes

Reviewer #2: Yes

5. Review Comments to the Author

Reviewer #1: Using a historical dataset, this study performed a clustering analysis utilizing

the sparse subspace clustering (SSC) method to classify 271 past societies

spread across globe, and found that five clusters contained in two

superclusters are best fit according to silhouette score.

I found it is very interesting to see archetypes of historical societies such

as Roman Kingdom through the clustering analysis.

Because I am not an expert of historical studies, below I note comments only for

the statistical methods. I think some of them are improper and require to be

appropriately revised.

In the manuscript, details for the method of DNN imputation is lacking.

Which software did you use (e.g. chainer and tensorflow)?

Which optimizer did you employ (e.g. momentum, adagrad, adadelta, etc.)?

Moreover, DNN has many parameters: How many layers? How many units in each

layer? How did you optimize hyperparameters? It would be nice if all the

details are revealed so that one can easily replicate the present study.

Lines 250-253: The authors wrote that PC1 computed by a PCA analysis using the

same dataset was included in the input dataset for the main clustering analysis,

and later the authors compared PC1 and the clustering result (Figs 2-6).

I felt that it sounds tautological. Because the clustering uses information

provided by PC1, PC1 and the clustering result is correlating as a matter of

course. Removing PC1 from the input dataset should be scientifically better.

Moreover, information contained in PC1 is contained in the other features

(because PC1 is a summary of them) and therefore PC1 is redundant information.

The SSC clustering should, in theory, reproduce the same result without PC1

information.

Line 252: The authors reported that PC1 contains only 61% of the total

variance. Why not considering to look at higher PCs? For example, it may be

interesting to draw Fig 2 in the PC1-PC2 plane and examine whether clusters 0

and 1 (3 and 4) are clearly separated.

Lines 309-310: "... find a clustering that minimizes the number of data points

with a negative silhouette coefficient." I wonder if this criterion is

statistically valid. There are a number of criteria to evaluate goodness of a

non-supervised clustering result such as MDL, BIC, information entropy, etc. It

could be more convincing to use one of them instead of the present one.

Lines 310-312: "... we finish by manually iterating over data points with a

negative silhouette to relocate them to the cluster where they have the highest

silhouette." It would be nice to describe this procedure in a computationally

reproducible way. If the authors can manually optimize the clustering result,

why the 'hyperopt' optimizer was not able to do so?

Around line 316: The authors may be able to evaluate the goodness of fit of the

clustering by means of a bootstrapping method. Generating a number of shuffled

dataset in each of which 271 cases are uniformly randomized for each feature,

performing SSC clustering for each shuffled dataset, computing silhouette

scores, and here we have a distribution of silhouette score if there were no

significantly clustered structure. Using this, it would be possible to test

whether or not the score 0.18 is significant.

Reviewer #2: This is a very interesting analysis of the data on social complexity variables published by the Seshat Databank. As the authors acknowledge, their results are largely confirmatory. However, it is very important that claims made by previous analyses of these data are tested by using different analysis techniques. Particularly interesting is a high degree of correspondence between the current results and previous results by Peregrine, based on a different data set. For this reason, my recommendation is to publish after a minor revision.

My recommendations on how to improve this article focus primarily on the presentation and the terminology issues, apart from the substantive issue 3.

1. In the title, "mathematically" seems to be the wrong word. Statistically? Computationally? A better word is needed.

2. "p-hacking" on p. 1: explain this term

3. On p. 5, eqn (1). Shouldn't time be exponentially discounted, similarly to space? Yes, it would require an additional parameter. But remember that the depth of history that Seshat presents varies between different areas. If there is no discounting of distant past, long series (e.g., Anatolia) will accumulate large x_0,i,t -- and that doesn't make sense.

4. I don't understand how the "edge case" arises -- how is it possible to have a numch of identical Y_i? Does that result form multiple reuse of the same data point?

5. I dislike "temperoculture" It evokes temperature, not time. Why not use the simpler polity-centuries?

6. Fig. 1 seems to be missing silhouettes.

7. Fig. 7: time spent in each clusters is a useful diagnostic. But it would also be useful to see if superclusters 1 and 2 are attractors in the sense that most trajectories go "in" rather than "out".

8. Finally, I dislike the juxtaposition of "Scale" vs. "Technology". The second is really misleading. Why not say simply "Non-Scale"?

6. PLOS authors have the option to publish the peer review history of their article (what does this mean?). If published, this will include your full peer review and any attached files.

Reviewer #1: No

Reviewer #2: Yes: Peter Turchin

---

## [Author Response · Author response to Decision Letter 0]

3 Mar 2020

Please see attached response to reviewers

---

## [Decision Letter · Decision Letter 1]

14 Apr 2020

PONE-D-19-30886R1

The two types of society: computationally revealing recurrent social formations and their evolutionary trajectories

PLOS ONE

Dear Mx. Miranda,

Thank you for submitting your manuscript to PLOS ONE. After careful consideration, we feel that it has merit but does not fully meet PLOS ONE’s publication criteria as it currently stands. Therefore, we invite you to submit a revised version of the manuscript that addresses the points raised during the review process.

We would appreciate receiving your revised manuscript by May 29 2020 11:59PM. To enhance the reproducibility of your results, we recommend that if applicable you deposit your laboratory protocols in protocols.io, where a protocol can be assigned its own identifier (DOI) such that it can be cited independently in the future. For instructions see: http://journals.plos.org/plosone/s/submission-guidelines#loc-laboratory-protocols

We look forward to receiving your revised manuscript.

Kind regards,

Joe Yuichiro Wakano

Academic Editor

PLOS ONE

Additional Editor Comments (if provided):

Both reviewers are happy with the revised manuscript. Reviewer 1 requests a very technical issue (version of software) to appear. Please provide the requested information. After this minor addition, the manuscript will be accepted.

Reviewers' comments:

Reviewer's Responses to Questions

**Comments to the Author**

1. If the authors have adequately addressed your comments raised in a previous round of review and you feel that this manuscript is now acceptable for publication, you may indicate that here to bypass the “Comments to the Author” section, enter your conflict of interest statement in the “Confidential to Editor” section, and submit your "Accept" recommendation.

Reviewer #1: All comments have been addressed

Reviewer #2: All comments have been addressed

2. Is the manuscript technically sound, and do the data support the conclusions?

Reviewer #1: Yes

Reviewer #2: Yes

3. Has the statistical analysis been performed appropriately and rigorously? 

Reviewer #1: Yes

Reviewer #2: Yes

4. Have the authors made all data underlying the findings in their manuscript fully available?

Reviewer #1: Yes

Reviewer #2: Yes

5. Is the manuscript presented in an intelligible fashion and written in standard English?

Reviewer #1: Yes

Reviewer #2: Yes

6. Review Comments to the Author

Reviewer #1: I am very satisfied with the revised manuscript, except one minor point:

datawig's version seems not described. As the default parameters may be changed in version bump, the authors should note the software version.

Reviewer #2: My concerns have been addressed in a satisfactory manner. This article makes a significant contribution to the literature on the evolution of social complexity, and therefore I favor publication.

7. PLOS authors have the option to publish the peer review history of their article (what does this mean?). If published, this will include your full peer review and any attached files.

Reviewer #1: No

Reviewer #2: Yes: Peter Turchin

---

## [Author Response · Author response to Decision Letter 1]

15 Apr 2020

Following the advice of Reviewer 1, we have included the version number for the datawig package.

---

## [Editor Report · Decision Letter 2]

20 Apr 2020

The two types of society: computationally revealing recurrent social formations and their evolutionary trajectories

PONE-D-19-30886R2

Dear Dr. Miranda,

We are pleased to inform you that your manuscript has been judged scientifically suitable for publication and will be formally accepted for publication once it complies with all outstanding technical requirements.

With kind regards,

Joe Yuichiro Wakano

Academic Editor

PLOS ONE
---

## [Editor Report · Acceptance letter]

28 Apr 2020

PONE-D-19-30886R2 

The two types of society: computationally revealing recurrent social formations and their evolutionary trajectories 

Dear Dr. Miranda:

I am pleased to inform you that your manuscript has been deemed suitable for publication in PLOS ONE. Congratulations! Your manuscript is now with our production department. 

With kind regards,

on behalf of

Dr. Joe Yuichiro Wakano 

Academic Editor

PLOS ONE